# SuperDialseg: A Large-scale Dataset for Supervised Dialogue Segmentation

**Junfeng Jiang**[†]    **Chengzhang Dong**[‡]    **Sadao Kurohashi**[‡§]    **Akiko Aizawa**[§†*]

[†]The University of Tokyo    [‡]Kyoto University    [§]National Institute of Informatics

jiangjf@is.s.u-tokyo.ac.jp
{dong@nlp.ist.i, kuro@i}.kyoto-u.ac.jp
aizawa@nii.ac.jp

## Abstract

Dialogue segmentation is a crucial task for dialogue systems allowing a better understanding of conversational texts. Despite recent progress in unsupervised dialogue segmentation methods, their performances are limited by the lack of explicit supervised signals for training. Furthermore, the precise definition of segmentation points in conversations still remains as a challenging problem, increasing the difficulty of collecting manual annotations. In this paper, we provide a feasible definition of dialogue segmentation points with the help of document-grounded dialogues and release a large-scale supervised dataset called **SuperDialseg**, containing 9,478 dialogues based on two prevalent document-grounded dialogue corpora, and also inherit their useful dialogue-related annotations. Moreover, we provide a benchmark including 18 models across five categories for the dialogue segmentation task with several proper evaluation metrics. Empirical studies show that supervised learning is extremely effective in in-domain datasets and models trained on SuperDialseg can achieve good generalization ability on out-of-domain data. Additionally, we also conducted human verification on the test set and the Kappa score confirmed the quality of our automatically constructed dataset. We believe our work is an important step forward in the field of dialogue segmentation. Our codes and data can be found from: https://github.com/Coldog2333/SuperDialseg.

## 1 Introduction

Recently, the importance of dialogue segmentation for dialogue systems has been highlighted in academia (Song et al., 2016; Xing and Carenini, 2021; Gao et al., 2023) and industry (Xia et al., 2022). Dialogue segmentation aims to divide a dialogue into several segments according to the discussed topics, and therefore benefits the dialogue understanding for various dialogue-related tasks, including dialogue summarization (Liu et al., 2019b; Chen and Yang, 2020; Feng et al., 2021b; Zhong et al., 2022), response generation (Xie et al., 2021), response selection (Xu et al., 2021), knowledge selection (Yang et al., 2022), and dialogue information extraction (Boufaden et al., 2001). When using Large Language Models (LLMs) as dialogue systems (e.g., ChatGPT[1]), dialogue segmentation methods can also help us to save tokens during long-term conversations while minimizing information loss. Figure 1 shows an example of dialogue segmentation.

Owing to the scarcity of supervised dialogue segmentation datasets, previous studies developed unsupervised methods (Song et al., 2016; Xu et al., 2021; Xing and Carenini, 2021) but easily reached the ceiling. Noting that models trained with large-scale supervised text segmentation datasets are effective for segmenting documents (Koshorek et al., 2018; Arnold et al., 2019; Barrow et al., 2020), however, Xie et al. (2021) experimented that applying them directly on dialogues were inadequate because dialogue has its own characteristics.

Previous studies showed that deep learning (DL) models usually perform better with supervised training signals and converge faster (Krizhevsky et al., 2017; Koshorek et al., 2018). Therefore, despite the expensive cost of manual annotation, previous works still attempted to collect some annotated datasets. However, these datasets are either small, containing only $300 \sim 600$ samples for training (Xie et al., 2021; Xia et al., 2022), or not open-sourced, due to the privacy concerns (Takanobu et al., 2018) and other issues (e.g., copyright issues from the industry (Xia et al., 2022)). Therefore, a large and publicly available supervised dataset is urgently needed. In addition, some studies on spoken dialogue systems provided manual annota-

---

[*]Corresponding Author

[1]https://openai.com/blog/chatgpt

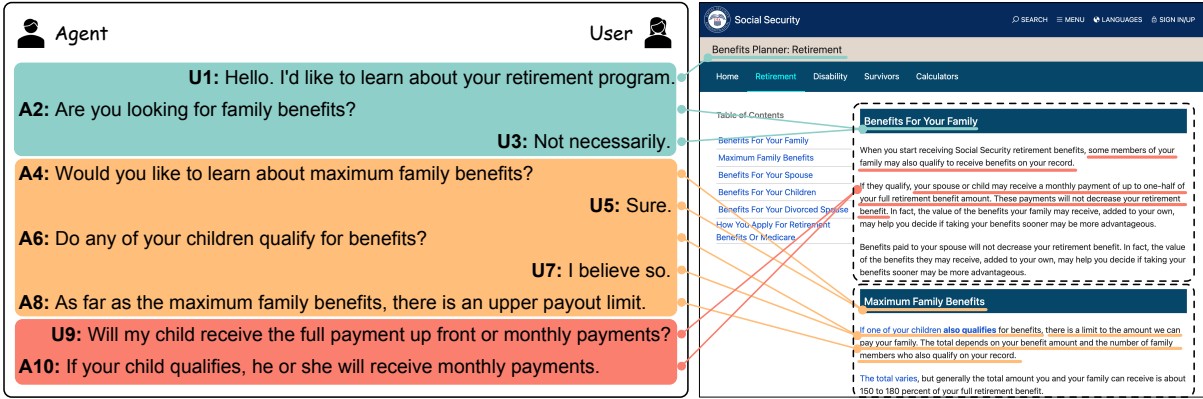

Figure 1: A document-grounded dialogue example selected from the doc2dial dataset (Feng et al., 2020). Different colors correspond to different dialogue segments.

tions on speech data (Janin et al., 2003; McCowan et al., 2005), but because of the noisy transcripts of spoken conversations, these corpora are also not suitable for training dialogue segmentation models.

Apart from the cost, another challenge for constructing a high-quality annotated dataset is the vague definition of segmentation points in dialogues. In contrast to documents, which have clear structures such as sections and paragraphs, segmenting dialogues is particularly confusing and difficult. Existing annotated datasets generally suffer from disagreement among annotators (Xie et al., 2021), resulting in relatively low Kappa scores[2] or boundary ambiguity (Xia et al., 2022). Inspired by Xing and Carenini (2021), we conjecture that the document-grounded dialogue system (DGDS) (Ma et al., 2020) can provide a criterion for defining the boundary of dialogue segments. Specifically, in a document-grounded dialogue, each utterance is grounded on a piece of relevant knowledge within a document, illustrating what speakers concentrate on or think about when talking. This information can reflect the discussing topic for each utterance. Based on this characteristic, we construct a supervised dialogue segmentation dataset, SuperDialseg, based on two DGDS datasets, containing 9,478 dialogues. Moreover, we inherit their rich dialogue-related annotations for broader applications.

Usually, dialogue segmentation is considered as a subtask that aids other downstream tasks, and thus, it lacks a standard and comprehensive benchmark for comparison. Therefore, in addition to proposing the supervised dataset, we conducted an extensive comparison including 18 models across five categories using appropriate evaluation metrics.

In summary, our contribution is threefold.

- We constructed a large-scale supervised dialogue segmentation dataset, SuperDialseg, from two DGDS corpora, inheriting useful dialogue-related annotations.

- We conducted empirical studies to show the good generalization ability of SuperDialseg and emphasized the importance of considering dialogue-specific characteristics for dialogue segmentation.

- We provide an extensive benchmark including 18 models across five categories. We believe it is essential and will be valuable for the dialogue research community.

## 2 Related Work

### 2.1 Text Segmentation

Before dialogue segmentation, text segmentation was already a promising task for analyzing document structures and semantics by segmenting a long document into several pieces or sections. To the best of our knowledge, cue phrases were used for text segmentation as early as 1983 (Brown and Yule., 1983; Grosz and Sidner, 1986; Hirschberg and Litman, 1993; Passonneau and Litman, 1997). Subsequently, Hearst (1997) proposed TextTiling to distinguish topic shifts based on the lexical co-occurrence. Choi (2000) and Glavaš et al. (2016) also proposed clustering-based and graph-based methods for text segmentation. Before large-scale datasets existed, these unsupervised methods performed relatively well but easily reached the ceiling. Facing this challenge, Koshorek et al. (2018)

---

[2]TIAGE's Kappa: 0.479 (0.41-0.6: moderate agreement)

collected 727k documents from Wikipedia and segmented them based on the natural document structures. With the Wiki727K dataset, they formulated text segmentation as a supervised learning problem and consequently led to many outstanding algorithms (Arnold et al., 2019; Barrow et al., 2020; Glavaš and Somasundaran, 2020; Gong et al., 2022). However, previous work (Xie et al., 2021) experimented that it is inadvisable to directly apply these studies to dialogue systems because dialogue possesses unique characteristics.

## 2.2 Document-grounded Dialogue System

Recently, limitations such as lacking common sense (Shang et al., 2015) and knowledge hallucination (Roller et al., 2021) have been identified in dialogue systems. To solve these issues, previous studies focused on extracting useful information from knowledge graphs (Jung et al., 2020), images (Das et al., 2017), and documents (Zhou et al., 2018; Dinan et al., 2019), yielding informative responses. Feng et al. (2020, 2021a) proposed two DGDS corpora with rich annotations, including role information, dialogue act, and grounding reference of each utterance, which enable knowledge identification learning and reveal how interlocutors think and seek information during the conversation.

## 2.3 Dialogue Segmentation

Unlike documents that have natural structures (e.g., titles, sections, and paragraphs), dialogue segmentation involves several challenges, including the vague definition of segmentation points, which increases the difficulty of data collection even for humans. Recently, Xia et al. (2022) manually collected 562 annotated samples to construct the CSTS dataset, and Xie et al. (2021) constructed the TIAGE dataset containing 300 samples for training. Not only was the size of annotated datasets too small for supervised learning, they also observed a similar issue: such datasets suffered from boundary ambiguity and noise frequently, even when the segmentation points were determined by human annotators. Zhang and Zhou (2019) claimed that they collected 4k manually labeled topic segments and automatically augmented them to be approximately 40k dialogues in total; however, until now, these annotated data are still not publicly available.

For evaluation, Xu et al. (2021) constructed the Dialseg711 dataset by randomly combining dialogues in different domains. Automatically generating large-scale datasets in this way seems plausi-

ble, but the dialogue flow around the segmentation point is not natural, resulting in semantic fractures. Though the dataset can still serve as an evaluation dataset, this characteristic can be interpreted as shortcuts (Geirhos et al., 2020), leading to poor generalization in real-world scenarios. Based on a prevalent DGDS dataset, doc2dial (Feng et al., 2020), Xing and Carenini (2021) used the training and validation sets to construct an evaluation dataset for dialogue segmentation but missed some crucial details. For example, as shown in Figure 1, the user begins the conversation with a general query referring to the document title but the following utterance from the agent refers to the first section. In this case, Xing and Carenini (2021) mistakenly assigned them to different segments, even though they are still discussing the same topic. Furthermore, most existing datasets neglected useful dialogue-related annotations like role information and dialogue act, while we inherit these annotations from the existing DGDS corpora to support future research. We will describe our dataset construction approach in Section 3.2. Table 1 summarizes the datasets mentioned above.

| Dataset | Reference | #Train | Ex-Anno | Open |
|---|---|---|---|---|
| SmartPhone | (Takanobu et al., 2018) | - | ✗ | ✗ |
| Clothing | | - | ✗ | ✗ |
| DAct | (Zhang and Zhou, 2019) | 20,000* | ✗ | ✗ |
| Sub | | 20,000* | ✗ | ✗ |
| Dialseg711 | (Xu et al., 2021) | - | ✗ | ✓ |
| CSM/doc2dial | (Xing and Carenini, 2021) | - | ✗ | ✓ |
| TIAGE | (Xie et al., 2021) | 300 | ✗ | ✓ |
| CSTS | (Xia et al., 2022) | 506 | Role | ✗ |
| **SuperDialseg** | **Ours** | **6,863** | Role,**DA** | ✓ |

Table 1: Summary of dialogue segmentation datasets. To the best of our knowledge, SuperDialseg is the largest open-sourced supervised dataset for dialogue segmentation. (Ex-Anno: Extra annotations, DA: Dialogue act)
*These data are augmented by random combinations.

## 3 The SuperDialseg Dataset

Documents have natural structures, whereas natural structures for dialogues are hard to define. In this paper, we utilize the well-defined document structure to define the dialogue structure.

## 3.1 Data Source

Doc2dial (Feng et al., 2020) and MultiDoc2Dial (Feng et al., 2021a) are two popular and high-quality document-grounded dialogue corpora. Compared to other DGDS datasets (Zhou et al., 2018; Dinan et al., 2019), they contain fine-grained

grounding references for each utterance, as well as additional dialogue-related annotations like dialogue acts and role information. Based on these annotations, we can construct a supervised dialogue segmentation dataset without human effort. Detailed information about doc2dial and MultiDoc2Dial can be found in Appendix B.

## 3.2 Data Construction Approach

Since doc2dial and MultiDoc2Dial provide the grounding texts for utterances, such that we can understand what the utterances are about according to the grounding references in documents. Take Figure 1 as an example. The user query '*Will my child receive the full payment up front or monthly payments*' refers to the information from the section 'Benefits For Your Family'. Then, the agent reads the document, seeking information from the same section, and provides a proper response. In this case, we know they are discussing a same topic because the subject of these utterances locates in the same section of the document. During this interaction in the DGDS scenario, we find that it provides an alignment between documents and dialogues. In this manner, we can define the segmentation points of dialogue based on the inherent structure of documents considering the grounding reference of each utterance. Furthermore, the grounding reference can reflect what interlocutors are thinking about when talking, which precisely matches the definition discussed in Section 1. Thus, this validates the rationality of our data collection method. In this paper, we focus on segmenting this type of dialogue, which is also dominant in recent ChatGPT-like dialogue systems (e.g., ChatPDF[3]). Surprisingly, our empirical studies indicate that models trained on our datasets can also perform well in other types of dialogue (i.e. non-document-grounded dialogue). We will discuss it later in Section 5.4.

Based on this idea, we determine that one utterance starts a new topic when its grounding reference originates from a different section in the grounded document compared to its previous utterance. Besides, there may be multiple grounding references for one utterance. In such a case, if all references of the current utterance are not located in the same section of any reference of the previous utterance, we regard it as the beginning of a new topic. Otherwise, these two utterances refer to the same topic. Note that MultiDoc2Dial has already

had document-level segmentation points, but our definition is more fine-grained and can describe the fine-grained topic shifts in conversations.

Furthermore, we carefully consider some details in the DGDS datasets. In the doc2dial and MultiDoc2Dial datasets, users sometimes ask questions that require agents' further confirmation, like '*Hello. I'd like to learn about your retirement program*' in Figure 1. However, in this case, though these queries and their following utterances are discussing the same topic, their grounding spans may not locate in the same section, leading to mistakes. Therefore, we corrected it as a non-segmentation point if the dialogue act of utterances from users is 'query condition' (the user needs further confirmation). Besides, we also removed the dialogues with meaningless utterances like 'null'/'Null' to further improve the quality of our supervised dataset.

Moreover, the dev/test sets of the DGDS datasets contain some unseen documents with respect to the training set. Consequently, some dialogues in the dev/test sets discuss unseen topics with respect to the training set. Therefore, following the official splitting can serve as an excellent testbed to develop dialogue segmentation models with high generalization ability. Algorithm 1 shows the procedure of determining the segmentation points for a document-grounded dialogue. Note that the $g_i$ is a set containing single/multiple grounding reference(s).

---

**Algorithm 1** Segment a dialogue

---

**Require:** $U = \{u_i\}_{i=0}^N$: utterances
**Require:** $G = \{g_i\}_{i=0}^N$: grounding spans
**Require:** $Sec(g)$: return the section IDs
**Require:** $R = \{r_i\}_{i=0}^N$: roles
**Require:** $DA = \{da_i\}_{i=0}^N$: dialogue acts
1: **for** $i = 1, \cdots, N$ **do**
2:    **if** $Sec(g_i) \cap Sec(g_i - 1) = \emptyset$ **then**
3:       $L(u_{i-1}) = 1$
4:    **end if**
5: **end for**
6: **for** $i = 0, \cdots, N$ **do**
7:    **if** $da_i ==$ query condition **and** $r_i ==$ user **then**
8:       $L(u_i) = 0$
9:    **end if**
10: **end for**
11: $L(u_N) = 1$
12: **return** $L$: List of segmentation points.

---

[3] https://www.chatpdf.com/

Besides, doc2dial and MultiDoc2Dial provide useful dialogue-related annotations about the role (user or agent) and dialogue act of each utterance (e.g., `respond solution`). We inherit them to enrich our supervised dialogue segmentation corpus. Intuitively, dialogue segments are characterized by specific patterns of roles and dialogue acts. For example, when users ask questions, the agents will provide answers or ask another question to confirm some details. The discussed topics are generally maintained until the problem is solved. Therefore, we hypothesize that introducing dialogue act information is helpful for dialogue segmentation. We conducted a pilot experiment on the relationship between the segmentation points and the dialogue acts, proving our hypothesis. Figure 2 illustrates the distribution of dialogue acts for all utterances and those serve as the segmentation points in the test set of SuperDialseg. We observe that most dialogue acts are highly related to the segmentation points. Specifically, utterances with dialogue acts DA-1, DA-5, DA-6, and DA-7 probably act as segmentation points, which indicates that a topic usually ends when solutions are provided, or the previous query has no solution. Therefore, these extra dialogue-related annotations are significant for the task of dialogue segmentation. We also conducted an analysis in Section 5.4.4 to further confirm our hypothesis.

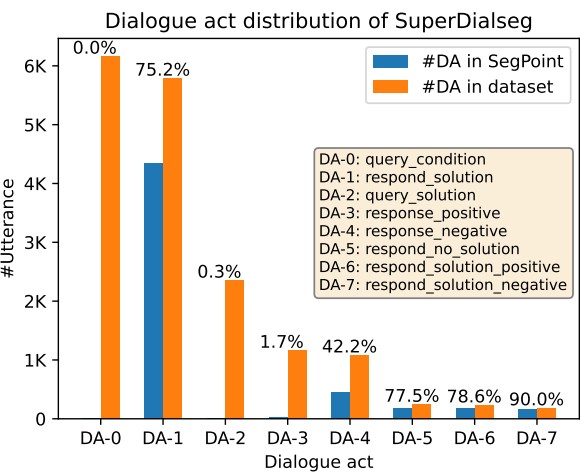

Figure 2: Distribution of dialogue acts for all utterances and for those at the segmentation points in the test set of SuperDialseg. The percentages represent the proportion of utterances where they serve as segmentation points among all utterances belonging to the same dialogue act.

Table 2 summarizes the statistics of the SuperDialseg dataset. Detailed distribution of the number

of segment is shown in Appendix C.

| Split | #Sample | #Token/Dial | | #Turn/Dial | | #Segment/Dial | |
|---|---|---|---|---|---|---|---|
| | | Aver | Max | Aver | Max | Aver | Max |
| Train | 6,863 | 239 | 646 | 13 | 24 | 4.25 | 10 |
| Valid | 1,305 | 231 | 553 | 13 | 20 | 4.27 | 9 |
| Test | 1,310 | 220 | 548 | 13 | 20 | 4.09 | 9 |

Table 2: Statistics of our SuperDialseg dataset.

## 3.3 Human Verification

To verify the quality of our proposed dataset, we conducted human verification to confirm the rationality of our data collection method. We randomly selected 100 dialogues from the test set of SuperDialseg and each dialogue was annotated by two human annotators. We collected the annotations from the crowd-sourcing platform, Amazon Mechanical Turk [4], and 20 valid workers participated in our annotation task. Each assignment contains 10 dialogues to be annotated.

Since it is a verification process, it should be conducted more carefully than collecting data for training. Therefore, we only allowed workers whose HIT approval rate is greater than 95% to access and also rejected some invalid results, for example, including too many consecutive segmentation points. Detailed rejection rules and the user interface of our verification task can be found in Appendix D. For each dialogue, we asked two unique workers to do the annotation and calculated the Cohen's Kappa scores between their annotations and the labels constructed by our data collection method. Finally, the Kappa score is 0.536, which is higher than TIAGE's Kappa score. Therefore, we can conclude that SuperDialseg has similar or even higher quality than the manually collected corpus (i.e., TIAGE) so that it can serve as a good training resource and testbed for the task of dialogue segmentation. Note that our automatic data collection method does not require human effort and can be easily scaled up as long as extra annotated document-grounded dialogue corpora are available.

## 4 Task definition

Dialogue Segmentation aims to segment a dialogue $D = \{U_1, U_2, ..., U_n\}$ into several pieces according to their discussed topics, where $U_i$ contains not only the utterance $u_i$, but also some properties including the role $r_i$ and the dialogue act $da_i$.

---

[4] https://www.mturk.com/

With the SuperDialseg dataset, we define it as a supervised learning task by labeling each utterance with $y = 1$ if it is the end of a segment, otherwise, $y = 0$.

## 5 Experiments

### 5.1 Datasets

**SuperDialseg** is our proposed dataset, containing 9,478 conversations with rich annotations. Details are provided in Section 3.

**TIAGE** (Xie et al., 2021) is a manually annotated dataset extended from PersonaChat (Zhang et al., 2018) comprising 500 samples, where 300, 100, and 100 samples are for the training, testing, and validation sets, respectively.

**DialSeg711** (Xu et al., 2021) is an evaluation dataset containing 711 multi-turn conversations, which are synthesized by combining single-topic dialogues sampled from the MultiWOZ (Budzianowski et al., 2018) and the Stanford Dialogue Dataset (Eric et al., 2017).

### 5.2 Comparison Methods

We construct an extensive benchmark involving 18 models across five categories, including random methods, unsupervised traditional methods, unsupervised DL-based methods, LLM-based methods, and supervised DL-based methods.

**Random Methods**. We design two random methods to show the bottom bound of the dialogue segmentation task. In an $n$-turn dialogue, we randomly select the number of segmentation points $c$ from $\{0, \cdots, n-1\}$. Then, Random places a segmentation point with the probability $\frac{c}{n}$, whereas Even distributes the segmentation boundaries evenly based on the interval $\lfloor \frac{n}{c} \rfloor$.

**Unsupervised Traditional Methods**. We select three representative algorithms in this category. BayesSeg (Eisenstein and Barzilay, 2008) is a Bayesian lexical cohesion segmenter. GraphSeg (Glavaš et al., 2016) segments by solving maximum clique problem of a semantic association graph with utterance nodes. TextTiling (Hearst, 1997) determines the segmentation point when the calculated depth score of a pair of adjacent utterances is beyond a given threshold.

**Unsupervised DL-based Methods**. Song et al. (2016) enhanced TextTiling by word embeddings. We use Glove embedding (Pennington et al., 2014) here and denote it as TextTiling+Glove.

TextTiling+[CLS] (Xu et al., 2021) further enhanced TextTiling with [CLS] representation from pre-trained BERT (Devlin et al., 2018). Xu et al. (2021) improved the TextTiling algorithm to be the GreedySeg algorithm to provide more accurate dialogue segments for response selection. Xing and Carenini (2021) designed TextTiling+NSP, which uses the next sentence prediction (NSP) score of BERT as the similarity between two adjacent utterances to enhance TextTiling, and also pre-trained a coherence scoring model (CSM) on large-scale dialogue corpora with 553k samples to compute the coherence score of two adjacent utterances instead. Koshorek et al. (2018) proposed an LSTM-based supervised text segmentation model called TextSeg. Since it is trained on the Wiki727K dataset without using annotated dialogue segmentation datasets, we regard it as an unsupervised method for dialogue segmentation, denoted as $\text{TextSeg}_{text}$.

**LLM-based Methods**. Recently, LLMs achieve remarkable success in various fields. We investigate how well they can segment dialogues by applying the InstructGPT (Ouyang et al., 2022) and ChatGPT based on simple prompt templates.

**Supervised DL-based Methods**. We design some strong baselines based on pre-trained language models (PLMs), including BERT, RoBERTa (Liu et al., 2019a), and TOD-BERT (Wu et al., 2020). Besides, we train TextSeg but using SuperDialseg or TIAGE dataset, denoted as $\text{TextSeg}_{dial}$. Xie et al. (2021) proposed RetroTS-T5, which is a generative model for dialogue segmentation.

We try our best to reproduce the reported performances in published papers with official open-sourced resources and our implementation. Most of the results are similar or even superior to those reported. Further details about all the methods and the implementation are provided in Appendix A for reproducing the results in our benchmark.

### 5.3 Evaluation Metrics

To evaluate the model performances, we use the following metrics: (1) $P_k$-error (Beeferman et al., 1999) checks whether the existences of segmentation points within a sliding window in predictions and references are coincident; (2) WindowDiff (WD) (Pevzner and Hearst, 2002) is similar to $P_k$, but it checks whether the numbers of segmentation points within a sliding window in predictions and references are the same; $P_k$ and WD compute soft errors of segmentation points within sliding

| Model | SuperDialseg | | | | | TIAGE | | | | | Dialseg711 | | | | |
|---|---|---|---|---|---|---|---|---|---|---|---|---|---|---|---|
| | Pk↓ | WD↓ | F1↑ | MAE↓ | Score↑ | Pk↓ | WD↓ | F1↑ | MAE↓ | Score↑ | Pk↓ | WD↓ | F1↑ | MAE↓ | Score↑ |
| **Without Any Annotated Corpus** | | | | | | | | | | | | | | | |
| Random | 0.494 | 0.649 | 0.266 | 3.819 | 0.347 | 0.526 | 0.664 | 0.237 | 4.793 | 0.321 | 0.533 | 0.714 | 0.204 | 9.760 | 0.290 |
| Even | 0.471 | 0.641 | 0.308 | 4.075 | 0.376 | 0.489 | 0.650 | 0.241 | 5.095 | 0.336 | 0.502 | 0.690 | 0.269 | 10.061 | 0.337 |
| BayesSeg | 0.433 | 0.593 | 0.438 | 2.934 | 0.463 | 0.486 | 0.571 | 0.366 | 2.870 | 0.419 | 0.306 | 0.350 | 0.556 | 2.127 | 0.614 |
| TextTiling | 0.441 | 0.453 | 0.388 | 1.056 | 0.471 | 0.469 | 0.488 | 0.204 | 1.420 | 0.363 | 0.470 | 0.493 | 0.245 | 1.550 | 0.382 |
| GraphSeg | 0.450 | 0.454 | 0.249 | 1.721 | 0.398 | 0.496 | 0.515 | 0.238 | 1.430 | 0.366 | 0.412 | 0.442 | 0.392 | 1.900 | 0.483 |
| TextTiling+Glove | 0.519 | 0.524 | 0.353 | 1.192 | 0.416 | 0.486 | 0.511 | 0.236 | 1.350 | 0.369 | 0.399 | 0.438 | 0.436 | 1.948 | 0.509 |
| TextTiling+[CLS] | 0.493 | 0.523 | 0.277 | 1.195 | 0.385 | 0.521 | 0.556 | 0.218 | 1.400 | 0.340 | 0.419 | 0.473 | 0.351 | 2.236 | 0.453 |
| GreedySeg | 0.490 | 0.494 | 0.365 | 1.490 | 0.437 | 0.490 | 0.506 | 0.181 | 1.520 | 0.341 | 0.381 | 0.410 | 0.445 | **1.419** | 0.525 |
| TextTiling+NSP | 0.512 | 0.521 | 0.208 | 1.747 | 0.346 | 0.425 | 0.439 | 0.285 | 1.800 | 0.426 | 0.347 | 0.360 | 0.347 | 1.969 | 0.497 |
| CSM | 0.462 | 0.467 | 0.381 | 1.261 | 0.458 | 0.400 | 0.420 | 0.427 | 1.370 | 0.509 | 0.278 | 0.302 | 0.610 | 1.450 | 0.660 |
| TextSeg$_{text}$ | 0.823 | 0.838 | 0.055 | 2.867 | 0.113 | 0.852 | 0.855 | 0.005 | 3.090 | 0.076 | 0.934 | 0.945 | 0.011 | 3.689 | 0.036 |
| InstructGPT | 0.365 | 0.377 | 0.529 | 1.255 | 0.579 | 0.510 | 0.552 | 0.280 | 1.930 | 0.375 | 0.410 | 0.465 | 0.515 | 3.838 | 0.539 |
| ChatGPT | 0.318 | 0.347 | 0.658 | 1.610 | 0.663 | 0.496 | 0.560 | 0.362 | 2.670 | 0.417 | 0.290 | 0.355 | **0.690**[†] | 3.423 | 0.684 |
| **Supervised Learning on TIAGE** | | | | | | | | | | | | | | | |
| TextSeg$_{dial}$ | 0.552 | 0.570 | 0.199 | 1.672 | 0.319 | 0.357 | 0.386 | 0.450 | 1.339 | 0.539 | 0.476 | 0.491 | 0.182 | 1.794 | 0.349 |
| BERT | 0.511 | 0.513 | 0.043 | 2.955 | 0.266 | 0.418 | 0.435 | 0.124 | 2.520 | 0.349 | 0.441 | 0.411 | 0.005 | 3.815 | 0.297 |
| RoBERTa | 0.434 | 0.436 | 0.276 | 2.262 | 0.420 | **0.265**[†] | **0.287**[†] | _0.572_ | **1.206** | **0.648**[†] | **0.197**[†] | **0.210**[†] | 0.650 | 1.325 | **0.723** |
| TOD-BERT | 0.505 | 0.514 | 0.087 | 2.447 | 0.289 | 0.449 | 0.469 | 0.128 | 2.071 | 0.335 | 0.437 | 0.463 | 0.072 | 3.708 | 0.311 |
| RetroTS-T5 | 0.504 | 0.505 | 0.095 | 2.867 | 0.295 | _0.280_ | _0.317_ | **0.576** | _1.260_ | _0.639_ | 0.331 | 0.352 | 0.303 | 1.442 | 0.481 |
| **Supervised Learning on SuperDialseg** | | | | | | | | | | | | | | | |
| TextSeg$_{dial}$ | _0.199_ | _0.204_ | _0.760_ | **0.813** | _0.779_ | 0.489 | 0.508 | 0.266 | 1.678 | 0.384 | 0.453 | 0.461 | 0.367 | 3.387 | 0.455 |
| BERT | 0.214 | 0.225 | 0.725 | 0.934 | 0.753 | 0.492 | 0.526 | 0.226 | 1.639 | 0.359 | 0.401 | 0.473 | 0.381 | 2.921 | 0.472 |
| RoBERTa | **0.185**[†] | **0.192**[‡] | **0.784**[†] | _0.816_ | **0.798**[†] | 0.401 | 0.418 | 0.373 | 1.495 | 0.482 | _0.241_ | _0.272_ | 0.660 | _1.460_ | _0.702_ |
| TOD-BERT | 0.220 | 0.233 | 0.740 | 0.910 | 0.757 | 0.505 | 0.537 | 0.256 | 1.622 | 0.367 | 0.361 | 0.454 | 0.534 | 3.438 | 0.563 |
| RetroTS-T5 | 0.227 | 0.237 | 0.733 | 1.005 | 0.751 | 0.415 | 0.439 | 0.354 | 1.500 | 0.463 | 0.321 | 0.378 | 0.493 | 2.041 | 0.572 |

Table 3: Performances on three datasets. Models in the first part do not access any annotated corpus. Models in the second part are trained on the TIAGE dataset and evaluated on the testing set of SuperDialseg, TIAGE, and Dialseg711. Models in the third part are similar to the second part's models but trained on SuperDialseg. The best and second-best performances are highlighted in bold and underlined, respectively. Different superscripts represent different significance levels (†: $p < 0.001$ (***), ‡: $p < 0.01$ (**)).

windows, and lower scores indicate better performances. (3) $F1$ score; (4) mean absolute error (MAE) calculates the absolute difference between the numbers of segmentation points in predictions and references; (5) By considering the soft errors and the $F1$ score simultaneously, we use the following score for convenient comparison:

$$Score = \frac{2 * F1 + (1 - P_k) + (1 - WD)}{4}, \quad (1)$$

suggested by the ICASSP2023 General Meeting Understanding and Generation Challenge (MUG).[5]

For the $P_k$ and WD, previous works adopted different lengths of sliding windows (Xu et al. (2021) used four, but Xing and Carenini (2021) used different settings), leading to an unfair comparison. Following Fournier (2013), we set the length of the sliding window as half of the average length of segments in a dialogue, which is more reasonable because each dialogue has a different average length of segments. In contrast to Xing and

Carenini (2021), we use $F1$ (binary) instead of $F1$ (macro) in the evaluation, because we only care about the segmentation points.

## 5.4 Experimental Results and Analysis

We conduct extensive experiments to demonstrate the effectiveness of our proposed dataset and reveal valuable insights of dialogue segmentation. Table 3 shows the results of our benchmark.

### 5.4.1 Supervised vs. Unsupervised

When trained on SuperDialseg, supervised methods outperform all unsupervised methods on SuperDialseg by large margins, and they surpass most of the unsupervised methods on TIAGE when trained on TIAGE, which highlights the crucial role of supervised signals in dialogue segmentation. Despite the domain gap, supervised methods exhibit strong generalization ability on out-of-domain data. For example, when trained on SuperDialseg, RoBERTa surpasses all unsupervised methods on Dialseg711 and all except CSM on TIAGE.

[5]https://2023.ieeeicassp.org/signal-processing-grand-challenges/

| Model | Train | SuperDialseg | | | | TIAGE | | | | Dialseg711 | | | |
|---|---|---|---|---|---|---|---|---|---|---|---|---|---|
| | | Pk↓ | WD↓ | F1↑ | Score↑ | Pk↓ | WD↓ | F1↑ | Score↑ | Pk↓ | WD↓ | F1↑ | Score↑ |
| BERT | TIAGE | 0.511 | 0.513 | 0.043 | 0.266 | **0.418**$^{†}$ | **0.435**$^{†}$ | 0.124 | **0.349** | 0.441 | **0.411** | 0.005 | 0.297 |
| TOD-BERT | | 0.505 | 0.514 | 0.087 | 0.289 | 0.449 | 0.469 | 0.128 | 0.335 | **0.437** | 0.463 | 0.072 | 0.311 |
| BERT | SuperDialseg-300 | **0.297**$^{†}$ | **0.316**$^{†}$ | **0.622**$^{†}$ | **0.658**$^{†}$ | 0.534 | 0.626 | **0.278** | **0.349** | 0.460 | 0.568 | 0.326 | **0.406**$^{‡}$ |
| TOD-BERT | | 0.370 | 0.392 | 0.541 | 0.580 | 0.541 | 0.643 | 0.265 | 0.336 | 0.480 | 0.622 | **0.349** | 0.399 |

Table 4: Comparison at the same scale of training data. The best performances are highlighted in bold. Different superscripts represent different significance levels (†: $p < 0.001$ (***), ‡: $p < 0.01$ (**)).

### 5.4.2 SuperDialseg vs. TIAGE

Although TIAGE is the first open-sourced supervised dataset for dialogue segmentation, it only has 300 training samples, which may not be sufficient for deep learning models. Our experiments show that trained on TIAGE, BERT and TOD-BERT perform poorly on Dialseg711 with low $F1$ scores, while they even perform worse than random methods on SuperDialseg. Additionally, we observe that when trained on TIAGE, RetroTS-T5 usually generates strange tokens such as '<extra_id_0> I' and 'Doar pentru', rather than 'negative'/'positive'. These observations reflect that small training sets generally result in unstable training processes.

Besides, we try to conduct a comparison between SuperDialseg and TIAGE on the same scale. We randomly selected 300/100 training/validation samples from SuperDialseg, denoted as SuperDialseg-300. Table 4 shows that though the performances are worse than those trained with the full training set, training with the subset of SuperDialseg is still more stable and performs relatively well on all datasets. We believe it is because SuperDialseg has more diverse patterns for dialogue segmentation. Furthermore, except for RoBERTa, other models trained on SuperDialseg outperform those trained on TIAGE when evaluating on Dialseg711, which proves the superiority of our proposed dataset.

### 5.4.3 DialogueSeg vs. TextSeg

It may seem plausible that models trained on large amounts of encyclopedic documents can be used to segment dialogues if we treat them as general texts. However, our results show that even though TextSeg$_{text}$ is trained with more than 100 times (w.r.t. SuperDialseg) and even 2,400 times (w.r.t. TIAGE) more data than TextSeg$_{dial}$, it performs significantly worse than TextSeg$_{dial}$ on all evaluation datasets. We attribute this to the fact that TextSeg$_{text}$ neglects the special characteristics of dialogues, and thus, is unsuitable for dia-

logue segmentation. For example, documents generally discuss a topic with many long sentences, whereas topics in conversations generally involve fewer relatively shorter utterances. Therefore, even though we have large-scale text segmentation corpora, learning only on documents is not sufficient for segmenting dialogues, which convinces the necessity of constructing large-scale supervised segmentation datasets specially for dialogues.

### 5.4.4 Effect of Dialogue-related Annotations

SuperDialseg provides additional dialogue-related annotations such as role information and dialogue acts. We conduct an analysis to show their effect on dialogue segmentation. We introduce role information and dialogue act by adding trainable embeddings to the inputs, denoted as the Multi-view RoBERTa (MVRoBERTa). Table 5 shows that MVRoBERTa outperforms the vanilla RoBERTa, which confirms that considering dialogue-related annotations is important for dialogue segmentation.

| Model | SuperDialseg | | | | |
|---|---|---|---|---|---|
| | Pk↓ | WD↓ | F1↑ | MAE↓ | Score↑ |
| RoBERTa | 0.185 | 0.192 | 0.784 | 0.816 | 0.798 |
| MVRoBERTa | **0.178**$^{*}$ | **0.185**$^{*}$ | **0.797**$^{*}$ | **0.792** | **0.808**$^{*}$ |

Table 5: Experiment for exploring the effect of dialogue-related annotations. (*: $p < 0.05$ (*))

### 5.4.5 Other Valuable Findings

CSM is a strong unsupervised method, as it achieves the best performance on TIAGE and Dialseg711 compared with other unsupervised methods, LLM-based methods, and supervised methods, demonstrating the potential of unsupervised approaches in this task. However, it does not perform consistently well on SuperDialseg. As previously discussed in Section 5.1, utterances in Dialseg711 are incoherent when topics shift, and dialogues in TIAGE 'rush into changing topics' as stated in Xie et al. (2021). Therefore, CSM, which mainly learns sentence coherence features and coarse topic

information, fails to segment real-world conversations with fine-grained topics. This observation strengthens the necessity of our SuperDialseg dataset for evaluating dialogue segmentation models properly. Surprisingly, ChatGPT, despite using a simple prompt, achieves the best performance on SuperDialseg and Dialseg711 compared with other methods without accessing annotated corpora. Moreover, LLM-based methods perform competitively on TIAGE. We believe with better prompt engineering, they can serve as stronger baselines. Additionally, BayesSeg performs relatively well compared to some DL-based methods, showing the importance of topic modeling in this task. We highlight this as a future direction in this area.

## 6 Conclusions

In this paper, we addressed the necessities and challenges of building a large-scale supervised dataset for dialogue segmentation. Based on the characteristics of DGDS, we constructed a large-scale supervised dataset called SuperDialseg, also inheriting rich annotations such as role information and dialogue act. Moreover, we confirmed the quality of our automatically constructed dataset through human verification. We then provided an extensive benchmark including 18 models across five categories on three dialogue segmentation datasets, including SuperDialseg, TIAGE, and Dialseg711.

To evidence the significance and superiority of our proposed supervised dataset, we conducted extensive analysis in various aspects. Empirical studies showed that supervised signals are essential for dialogue segmentation and models trained on SuperDialseg can also be generalized to out-of-domain datasets. Based on the comparison between the existing dataset (i.e. TIAGE) and SuperDialseg on the same scale, we observed that models trained on SuperDialseg had a steady training process and performed well on all datasets, probably due to its diversity in dialogue segmentation patterns, therefore, it is suitable for this task. Furthermore, the comparison between RoBERTa and MVRoBERTa highlighted the importance of dialogue-related annotations previously overlooked in this task.

Additionally, we also provided some insights for dialogue segmentation, helping the future development in this field. Moreover, our dataset and codes are publicly available to support future research.[6]

## Limitations

In this study, we notice that our proposed method of introducing dialogue-related information including role information and dialogue act (i.e., MVRoBERTa) performs well but the way of exploiting these annotations is simple. We believe there is much room for improvement. Therefore, we need to find a more proper and effective way to make full use of the dialogue-related information, expanding the potential applications of supervised dialogue segmentation models.

Besides, our dataset is only available for training and evaluating English dialogue segmentation models. Recently, Fu et al. (2022) proposed a Chinese DGDS dataset with fine-grained grounding reference annotations, which can also be used to construct another supervised dataset for Chinese dialogue segmentation. We leave it as a future work.

## Ethics Statement

In this paper, we ensured that we adhered to ethical standards by thoroughly reviewing the copyright of two involved DGDS datasets, the doc2dial and the MultiDoc2Dial. We constructed the SuperDialseg dataset in compliance with their licenses (Apache License 2.0).[7] We took care of the rights of the creators and publishers of these datasets and added citations properly.

As for the process of human verification, we collected the data from Amazon Mechanical Turk. Each assignment cost the workers 20 minutes on average. Each worker was well paid with $2.5 after completing one assignment, namely, the hourly wage is $7.5, which is higher than the federal minimum hourly wage in the U.S. ($7.25).

## Acknowledgements

This work was supported by JST SPRING, Grant Number JPMJSP2108.

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

## A Details for Reproduction

### A.1 Comparison Methods

To guarantee a fair comparison in our benchmark, we use the open-sourced codes and pre-trained models of the comparison methods as much as possible to evaluate their performances. However, some additional details are required to reproduce their reported performances. Therefore, we conduct a similar level of hyperparameter tuning for each model to obtain similar or better results than those reported in previously published papers.

For the GraphSeg, we set the value of the relatedness threshold as 0.5 and the minimum segmentation sizes as 3 to obtain a better performance than that reported in Xing and Carenini (2021). For TextTiling+Glove, we used the version that was pre-trained with 42 billion tokens of web data from Common Crawl[8]. For GreedySeg and CSM, we corrected some inconsistencies in their open-sourced codes with respect to their original published papers and obtained better performances than their reported results.

For the result of LLM-based methods, we used the OpenAI API[9] with the best model `text-davinci-003` for InstructGPT and the recent model `gpt-3.5-turbo-0613` for ChatGPT in June 2023. The maximum number of completion tokens was 512, and the temperature was set to 0. We used the definition from our paper as described in Section 4 as the task instruction, together with the dialogue input, and an output example specifying the output format to form the input prompt. Table 6 shows our constructed prompt templates for the dialogue segmentation task.

For the PLM-based supervised methods, we experimented with the hierarchical architectures but did not achieve satisfactory results. Therefore, we followed the framework of Cohan et al. (2019) to concatenate all the utterances as a long sequence as input and use the [SEP]/ tokens at the end of each utterance except for TOD-BERT to perform the classification. Note that there are two separate tokens  in the original implementation of RoBERTa, we use the first  token to do classification. For TOD-BERT, each utterance has a special token representing its role, so we use the special role tokens [sys] and [usr] to perform classification. The maximum number of tokens in an utterance is 25. The size of the utterance-level sliding window during training and inference is $|T| = 20$, and the stride is 1.

Other implementations that are not specified follow the official codes and their papers with default hyperparameters.

### A.2 Implementation Details

We implemented all the models using PyTorch (Paszke et al., 2019) and downloaded model weights of PLMs from huggingface Transformers (Wolf et al., 2020). We also utilized Pytorch Light-

ning [10] to build the deep learning pipeline in our experiments. The AdamW (Loshchilov and Hutter, 2018) optimizer was used to optimize models' parameters with a learning rate of 1e-5 and a batch size of 8. L2-regularization with weight decay of 1e-3 was also applied. We trained our supervised models with 20 epochs for SuperDialseg and 40 epochs for TIAGE and SuperDialseg-300, respectively. We employed early stopping when the calculated score (metric (5) in Section 5.3) did not improve for half of the total number of epochs. For RetroTS-T5, we trained the generative T5 for 3 epochs on SuperDialseg and 40 epochs on TIAGE.

All experiments were conducted on a single A100 80GB GPU. Except for RetroTS-T5, all experiments took no longer than 1 hour to finish, whereas training RetroTS-T5 on SuperDialseg required approximately six hours. To reduce experimental error, we did experiments in several times and reported the average performances in this paper. T-test was also conducted to confirm the significance of the comparison.

## B  Detailed Information for Data Source

In real-world applications, the demand for guiding end users to seek relevant information and complete tasks is rapidly increasing. Feng et al. (2020) proposed a goal-oriented document-grounded dialogue dataset, called doc2dial, to help this line of research. To avoid additional noise from the post-hoc human annotations of dialogue data (Geertzen and Bunt, 2006), they designed the dialogue flows in advance. A dialogue flow is a series of interactions between agents and users, where each turn contains a dialogue scene that consists of dialogue acts, role information, and grounding contents from a given document. Specifically, by varying these three factors that are further constrained by the semantic graph extracted from documents and dialogue history, they dynamically built diverse dialogue flows turn-by-turn. Then, they asked crowdsourcing annotators to create the utterances for each dialogue scene within a dialogue flow. It should be noted that although the dialogue flows were generated automatically, the crowdsourcing annotators would reject some unnatural dialogue flows if they found any dialogue turn that was not feasible to write an utterance. Moreover, each dialogue was created based on a unique dialogue flow.

Usually, goal-oriented information-seeking con-

---

[8]https://nlp.stanford.edu/projects/glove/
[9]https://openai.com/api/

[10]https://pytorch-lightning.readthedocs.io/

versations involved multiple documents. Based on the doc2dial dataset, Feng et al. (2021a) randomly combined sub-dialogues from doc2dial which ground on different documents to construct the MultiDoc2Dial dataset. Therefore, each utterance only grounds on a single document. To maintain the natural dialogue flow after the combination, they only split a sub-dialogue after an agent turn with dialogue act as 'responding with an answer' while the next turn is not 'asking a follow-up question'. Besides, they also recruited crowdsourcing annotators to rewrite some unnatural utterances if the original utterances from doc2dial are inappropriate after combination.

Further details about the data collection methods for doc2dial and MultiDoc2Dial can be found in the original papers (Feng et al., 2020, 2021a).

## C    Detailed Information of SuperDialseg

Figure 3 shows the detailed distribution of the number of segments in the test set of SuperDialseg.

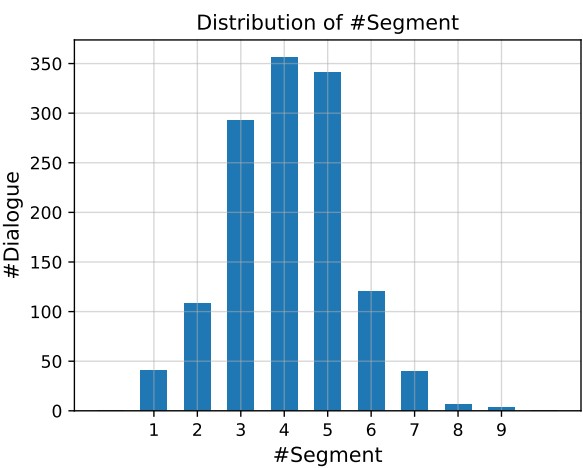

Figure 3: Distribution of #Segment in test set of Super-Dialseg.

## D    Detailed Information about Human Verification

Figure 4 shows the user interface of our human verification task. We randomly sampled 100 samples from the test set of SuperDialseg for human verification. For the sake of convenience in designing the user interface, we only selected those dialogues with less than 15 utterances.

We set up a series of constraints to ensure the quality of collected human verification data. First of all, we set up a constraint that only those workers whose HIT approval rate for all requesters' HITs is greater than 95% can access our task, which ensures the quality of our recruited annotators. However, we still observed some bad annotations. Therefore, we rejected those assignments if they satisfied any of the following conditions:

- The last utterance was not annotated as a segmentation point, indicating that the annotator did not read our given instructions carefully or did not fully understand our task.

- The 'N/A' utterance was annotated as a segmentation point, indicating that the annotator did not read our given instructions carefully.

- The example was modified, indicating that the annotator did not read our given instructions carefully.

- An annotation had more than three consecutive segmentation points, which is impossible in our cases.

- All the annotations remained in default.

Besides, we rejected those workers and would not approve any annotations from them ever if they satisfied any of the following conditions, because they were probably spam users:

- One submitted an assignment that had more than 10 consecutive segmentation points.

- One submitted an assignment whose all the annotations remained in default.

We hope that our practical experience in collecting verification data can also be helpful for future researchers who want to collect annotated dialogue segmentation data by crowdsourcing.

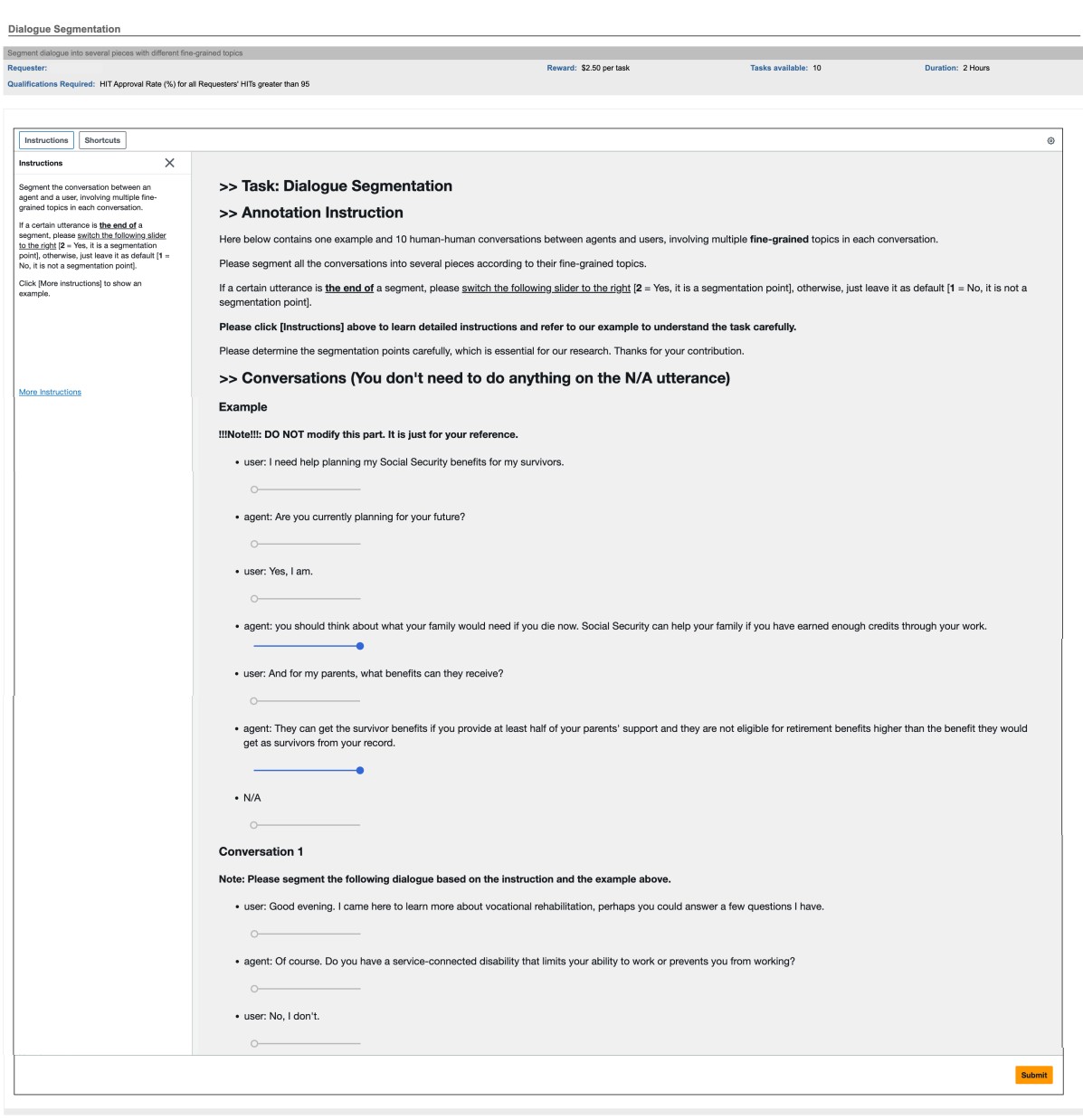

Figure 4: User interface for human verification task on Amazon Mechanical Turk.

**Prompt template for InstructGPT** (`text-davinci-003`)

\#Task Instruction
```
Dialogue Segmentation aims to segment a dialogue D = {U1, U2, ..., Un} into
several parts according to their discussed topics.
```
\# Input
```
Here is the given dialogue D:
U1: [Utterance 1]
U2: [Utterance 2]
...
```
Un: [Utterance $n$]
\# Output format
```
Segment D into several parts according to their discussing topics.
Output format: Part i: Ua-Ub
=====
Output example:
Part 1: U1-U4
Part 2: U5-U6
=====
```
\# Output
```
Output of the dialogue segmentation task:
```

**Prompt template for ChatGPT** (`gpt-3.5-turbo-0613`)

\# System message
```
You are a helpful assistance to segment give dialogues.
Please follow the output format.
DO NOT explain.
```
\# Task Instruction
```
Dialogue Segmentation aims to segment a dialogue D = {U1, U2, ..., Un} into
several parts according to their discussed topics.
Please help me to segment the following dialogue:
```
\# Input
```
U1: [Utterance 1]
U2: [Utterance 2]
...
```
Un: [Utterance $n$]
\# Output format
```
Segment D into several parts according to their discussing topics.
Output format: Part i: Ua-Ub
=====
Output example:
Part 1: U1-U4
Part 2: U5-U6
=====
```
\# Output
```
Output of the dialogue segmentation task:
```

Table 6: Prompt templates for dialogue segmentation task.