# OpenReview forum: "SuperDialseg: A Large-scale Dataset for Supervised Dialogue Segmentation"
_EMNLP/2023/Conference — EMNLP 2023 Main_

### Official Review · Reviewer_mUQu · 2023-08-01

**Soundness:** 3

**Excitement:**

3: Ambivalent: It has merits (e.g., it reports state-of-the-art results, the idea is nice), but there are key weaknesses (e.g., it describes incremental work), and it can significantly benefit from another round of revision. However, I won't object to accepting it if my co-reviewers champion it.

**Paper Topic And Main Contributions:**

This paper works on the dialogue segmentation problem.
The main contributions are: (1) constructing a large-scale dialogue segmentation dataset based on two existing document-grounded dialogue corpora, and (2) providing benchmark performance of 18 models in dialogue segmentation tasks.

**Questions For The Authors:**

Question A: could you provide a formal definition/pseudo code for your dialogue segmentation method?

Question B: the human verification part is confusing, why not directly ask the human to accept or reject your segmentation annotation?

**Reasons To Accept:**

1. Extensive benchmark experiments, which show that the supervised signals in the proposed dataset can greatly improve the model performance in dialogue segmentation.

**Reasons To Reject:**

1. Unclear writing of 3.2 data construction approach, lack of formal definition of the proposed dialogue segmentation annotation method.

**Reproducibility:**

3: Could reproduce the results with some difficulty. The settings of parameters are underspecified or subjectively determined; the training/evaluation data are not widely available.

**Reviewer Confidence:**

3: Pretty sure, but there's a chance I missed something. Although I have a good feel for this area in general, I did not carefully check the paper's details, e.g., the math, experimental design, or novelty.

---

> ### Author Rebuttal · Authors · 2023-08-28
>
> Thank you for your valuable feedback! We are glad to provide further details in response to your concerns about our paper.
>
> ### Question A: Could you provide a formal definition/pseudo code for your dialogue segmentation method?
> **Answer:** Yes, we are glad to do that. In Section 3.2, firstly, we introduced our motivations and the rationality of our method in Lines 236-266. Secondly, we introduced how to segment a dialogue in Lines 267-303. Thirdly, we discussed other dialogue-related information derived from the original DGDS corpora in the rest of this section.
>
> The whole data construction method can be separated into three parts: segmenting dialogues, data cleaning, and splitting them into train/val/test subsets. In segmenting dialogues, we 1) segmented dialogues based on the grounding spans, and 2) corrected segmentation errors based on dialogue acts. In data cleaning, we removed some bad dialogues. All the stages were done automatically based on designed rules.
>
> Here is a pseudo-code for explaining how to segment a dialogue (the first part of the whole process) for your reference:
>
> ```
> Algorithm: Segment a document-grounded dialogue (DGD)
> —
> Require: U={u_0,u_1,...,u_N}: utterances of a DGD.
> Require: G={g_0,g_1,...,g_N}: grounding spans of each utterance, g_i is a set containing single/multiple grounding reference(s).
> Require: Sec(g): return the section IDs of a set of grounding spans
> Require: R={r_0, r_1,...,r_N}: roles of each utterance
> Require: DA={da_0, da_1,...,da_N}: dialogue acts of each utterance
> —
> 1:  for i=1,...,N do
> 2:      if Sec(g_i) ∩ Sec(g_i-1) = ∅ do  # if they refer to different sections
> 3:          L(u_i-1)=1  # A new topic begins, so the previous utterance is the end of a segment.
> 4:      end if
> 5:  end for
> 6:  for i=0,1,...,N do   # correct segmentation errors
> 7:      if da_i == 'query condition' and r_i == 'user' do
> 8:          L(u_i)=0
> 9:      end if
> 10: end for
> 11: L(u_N)=1  # The last utterance naturally locates at the end of a segment.
> —
> Output: L: List of segmentation points.
> ```
>
> We also included the code for constructing SuperDialseg from doc2dial and MultiDoc2dial datasets in our supplementary materials. You can find the code from `supplementary/code/scripts/make_superdialseg.py`. And thanks for your comment, we will include this algorithm in our final version for future readers to understand our method more conveniently.
>
>
> ### Question B: The human verification part is confusing, why not directly ask the human to accept or reject your segmentation annotation?
> **Answer:** As for the process of our human verification, we have four reasons to do that.
> 1. Annotation for dialogue segmentation is not the same as that for some binary classification tasks, like sentiment analysis and response quality evaluation. Annotators need to determine the segmentation point for each utterance considering the whole conversation at the same time. Therefore, asking them to determine whether all the segmentation points are correct or not may not be applicable in dialogue segmentation.
> 2. In our human verification process, we intended to obtain unbiased and natural judgments as much as possible. So we asked them to directly segment a dialogue as usual instead of asking them to accept or reject our segmentation annotation. If we provided the annotations constructed by our method in advance, it would introduce some biases (so-called Leading Bias) that may affect their judgments. For example, at first glance, they may think it ‘acceptable’/‘plausible’ and therefore exaggerate the accuracy, resulting in a misleading verification result that is not sufficiently objective and rational.
> 3. Nowadays, collecting high-quality data from AMT is not as easy as before. If we only asked annotators to accept or reject our segmentation annotation, it would be hard for us to identify the spam annotations. Detailed information on how we control the quality of the human verification process can be found in Appendix D.
> 4. The annotated samples from human can also be reused for future research (e.g., to understand the differences in annotations from different annotators).

---

### Official Review · Reviewer_EsRS · 2023-08-05

**Soundness:** 4

**Excitement:**

4: Strong: This paper deepens the understanding of some phenomenon or lowers the barriers to an existing research direction.

**Paper Topic And Main Contributions:**

In the dialogue segment task, it is an open issue that the definition of segmentation point is ambiguous and that there is insufficient publicly available data to train supervised learning-based models. This paper proposes a feasible definition referring to document-grounded dialogue and introduces SuperDialseg, a large dataset (approximately 1K) for supervised dialogue segment tasks based on the definitions. The novel definition of segmentation point is highly agreed upon among human annotators (Section 2.3), so it seems a practical criterion. The SuperDialseg dataset is about 10x larger than the previous dataset and of equal or higher quality (Section 3). Along with SuperDialseg, the benchmarks include 18 models across five categories of dialogue segmentation tasks are also provided. This work has the potential to contribute to the development of future dialogue segmentation research.

**Reasons To Accept:**

- SuperDialseg dataset and benchmarks that includes modern archtecture e.g. BERT- and GPT-based models are useful for future research. These will be made available after acceptance.
- The human verification to check the quality of the created data is well-designed and reliable. Implementation of the verification details is also carefully explained and easy to understand.
- The analysis is substantial (Section 5.4), and the characteristics and availability of SuperDialseg are well explained.

**Reasons To Reject:**

- N/A

**Reproducibility:**

4: Could mostly reproduce the results, but there may be some variation because of sample variance or minor variations in their interpretation of the protocol or method.

**Reviewer Confidence:**

3: Pretty sure, but there's a chance I missed something. Although I have a good feel for this area in general, I did not carefully check the paper's details, e.g., the math, experimental design, or novelty.

---

> ### Author Rebuttal · Authors · 2023-08-28
>
> Thank you for your interest in our work! Our dataset, namely, SuperDialseg, and other involved preprocessed datasets (i.e. TIAGE and DialSeg711) will be publicly available immediately if our work is qualified as accepted. Our codes will also be open-sourced for this field of research. We hope our work can serve as a useful resource to develop better dialogue segmentation algorithms.

---

### Official Review · Reviewer_pN2F · 2023-08-07

**Soundness:** 4

**Excitement:**

4: Strong: This paper deepens the understanding of some phenomenon or lowers the barriers to an existing research direction.

**Paper Topic And Main Contributions:**

This paper presents a new dataset, SuperDialSeg, for the task of dialogue segmentation. Built upon the two document-grounding datasets, doc2dial and multidoc2dial, the authors leverage their existing annotations of the relationship between dialogue utterance and document sentence to derive the dialogue topic boundaries, with some dialogue characteristic considered and human verification on a small portion conducted. Extensive experiments are conducted on three datasets, SuperDialSeg, TIAGE, Dialseg711, with 18 different methods for comparison, including unsupervised, supervised models and LLMs.


**Reasons To Accept:**

- A useful large-scale corpus for dialogue segmentation is proposed, with a simple but effective annotation heuristic. This would potentially benefit dialogue related tasks.
- The paper is well-written, structured and motivated.
- Thorough experiments across three datasets and 18 methods are conducted, providing useful findings to the community.


**Reasons To Reject:**

Although some caveats are considered (e.g., confirmation dialogue act in Section 3.2) and human verification on a small portion (100 test dialogues) is conducted, the quality of the derived segmentation labels is bounded to that of annotations of the two DGDS corpora. The original annotation noise could be inherited and affects the segment boundary decisions.


**Reproducibility:**

4: Could mostly reproduce the results, but there may be some variation because of sample variance or minor variations in their interpretation of the protocol or method.

**Reviewer Confidence:**

4: Quite sure. I tried to check the important points carefully. It's unlikely, though conceivable, that I missed something that should affect my ratings.

---

> ### Author Rebuttal · Authors · 2023-08-28
>
> Thanks for your interest in our work! We also hope that SuperDialseg can contribute to the NLP community. And we believe that the following explanation will help to alleviate some of your concerns.
>
> We agree that the quality of the original DGDS corpora will affect the quality of SuperDialseg. But we would like to point out that the two DGDS datasets we chose are still of high quality and were used as shared tasks held in ACL 2021 and ACL 2022 for evaluating many document-grounded dialogue systems. Here are the home pages of these two datasets (https://doc2dial.github.io/ ; https://doc2dial.github.io/multidoc2dial/) respectively for your reference.
>
> As for some unexpected errors in these datasets, we also conducted multiple strategies to resolve this issue. For example, as stated in Lines 293-295, we removed the whole dialogue if there were meaningless utterances. We also corrected segmentation errors based on their dialogue act annotations, as stated in Lines 290-293. With these efforts, we believe our SuperDialseg can maintain a high quality for future research.

---

### Meta-Review · Area_Chair_R7cn · 2023-09-15

**Recommendation:** 5

**Metareview:**

The paper provides a new (supervised) dataset for dialogue segmentation.

**Pros**: All reviewers agree the dataset will be a useful resource to the community. Validation of the dataset is considered to be appropriate, using some human-verification used to verify the "simple but effective" annotation heuristic. Further, the motivations for the dataset are also well validated (the proposed supervisory signals add benefit). Indeed, reviewers note the benchmark experiments as being substantial and relevant to the community (3 benchmark datasets, 18 methods).

**Cons**: One reviewer requests missing details, while another points out that deriving supervisory signals from an existing datasets annotations (i.e., proxies) can incur noise, which will be compounded with the noise from the annotation heuristic. Authors appear to address these concerns during the rebuttal period.

---

### Decision · Program_Chairs · 2023-10-07

**Decision:**

Accept-Main

**Comment:**

The paper provides a new (supervised) dataset for dialogue segmentation.

**Pros**: All reviewers agree the dataset will be a useful resource to the community. Validation of the dataset is considered to be appropriate, using some human-verification used to verify the "simple but effective" annotation heuristic. Further, the motivations for the dataset are also well validated (the proposed supervisory signals add benefit). Indeed, reviewers note the benchmark experiments as being substantial and relevant to the community (3 benchmark datasets, 18 methods).

**Cons**: One reviewer requests missing details, while another points out that deriving supervisory signals from an existing datasets annotations (i.e., proxies) can incur noise, which will be compounded with the noise from the annotation heuristic. Authors appear to address these concerns during the rebuttal period.